# Learning Markerless Robot-Depth Camera Calibration and End-Effector Pose Estimation

**Bugra C. Sefercik**
CE Department / KUIS AI Center
Koc University, Turkiye
bsefercik@ku.edu.tr

**Baris Akgun**
CE Department / KUIS AI Center
Koc University, Turkiye
baakgun@ku.edu.tr

**Abstract:** Traditional approaches to extrinsic calibration use fiducial markers and learning-based approaches rely heavily on simulation data. In this work, we present a learning-based markerless extrinsic calibration system that uses a depth camera and does not rely on simulation data. We learn models for end-effector (EE) segmentation, single-frame rotation prediction and keypoint detection, from automatically generated real-world data. We use a transformation trick to get EE pose estimates from rotation predictions and a matching algorithm to get EE pose estimates from keypoint predictions. We further utilize the iterative closest point algorithm, multiple-frames, filtering and outlier detection to increase calibration robustness. Our evaluations with training data from multiple camera poses and test data from previously unseen poses give sub-centimeter and sub-deciradian average calibration and pose estimation errors. We also show that a carefully selected single training pose gives comparable results.

**Keywords:** Camera Calibration, Pose Estimation, Perception

## 1 Introduction

Camera to robot calibration is an important step in many robotic applications. Keeping the correct calibration is a challenge for robots in dynamic and uncontrolled environments, even if the robot and the camera are meant to be static. Multiple factors alter this calibration such as re-positioning the robot/camera for better workspace coverage or for different applications (e.g. research projects), inadvertently moving them for cleaning, people bumping into them, wear-and-tear and backlash on low-cost fixtures, etc. As such, calibration is a needed but time-consuming process. At the very least, calibration needs to be checked before an application. This is an all-too-real issue for robotics researchers and is getting more widespread as cage-free robot arms and mobile manipulators become more common.

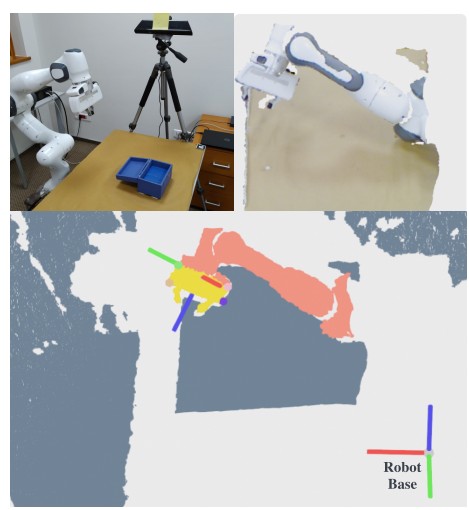

Figure 1: Top-left: experimental setup, top-right: sample frame from Kinect V1, bottom: segmentation, keypoints and pose prediction

Traditional extrinsic calibration approaches rely on fiducial markers. Checkerboard patterns and augmented reality (AR) tags are commonly used in research settings. Adding markers is error-prone due to intrinsic calibration errors, sensor noise, robot-to-marker fixture quality (which introduces transformation errors), etc. High precision systems employ precision machined fixtures and active markers (e.g. LEDs, reflectors+light-sources) which are costly. Thus, using markers is either noisy or expensive.

In this work, we develop a system that can handle the extrinsic calibration between a depth camera and a robot arm without additional hardware, markers or simulation. We utilize recent developments

6th Conference on Robot Learning (CoRL 2022), Auckland, New Zealand.

in deep learning models for point clouds in combination with the ICP algorithm. Our system collects its own data, negating the need for manual labeling, only requiring an initial calibration effort. Two side benefits of our system are that it provides high-quality semantic segmentation information about the background, end-effector (EE) and the rest of the robot, and estimates the EE pose with low error.

There are other learning-based approaches for markerless calibration for both eye-to-hand calibration [1, 2, 3] and eye-in-hand calibration, [4] that utilize colored images and synthetic/simulation data. Our work differs by exclusively using real-world data, negating the need for simulation, and depth data which readily enables the usage of the ICP algorithm for high performance.

The main contributions of our system are; (1) high performance (average test errors of $0.74cm$ and $1.69°$) markerless extrinsic calibration, (2) high performance (average test errors of $1.0cm$ and $2.74°$) EE pose estimation wrt. camera, and (3) automatic real-life data collection methodology.

## 2   Related Work

Calculating the transformation between a robot's base and a camera, usually called extrinsic calibration, is a decades old challenge [5]. Conventional techniques use fiducial markers to estimate camera to robot transformations. Most popular approaches stick a marker on the robot's EE [6, 7, 8] or other relevant parts [9, 10, 11] and collect data from multiple robot joint angle configurations to be used in offline calibration calculation. Major disadvantages of using such markers include wear-and-tear, distance dependent camera noise [12], fixture backlash and build quality, etc. There are also single frame methods that use fiducial markers to estimate the EE pose [13, 14] in real-time. These methods start to fail when the reference objects are not as visible enough as anticipated. In this work, we aim to develop a markerless approach to handle extrinsic calibration.

The advent of deep learning brought developments in areas that are closely related to EE pose estimation such as 6-DoF object pose estimation [15, 16] and keypoint detection [17, 18]. Several methods have produced successful results on robot pose estimation from a single frame. Lambrecht [1] and Lee et al. [2] learn to estimate keypoints of the entire robot arm from RGB image trained with newly created real world data and synthetic data, respectively, then to predict the robot pose combining keypoint data with forward kinematics information. Similarly, Labbé et al. [3] learns to predict robot pose and joint angles from RGB images via iterative CAD to image matching using a synthetic dataset [2]. Our system diverges from these methods in the following areas: (1) they require majority of the robot arm to be visible while our only requirement is to see the EE in the frame (2) our method uses depth data whereas others rely on Perspective-n-Point algorithm to compute affine transformations since they utilize 2D images, (3) we can readily apply the ICP algorithm to refine our pose estimates due to the depth data usage, and (4) we do not require any synthetic/simulation data and only use real-world data. It can be argued that we are solving a simpler problem (only EE and depth data) in exchange for higher calibration quality.

Hand-in-eye calibration, where the camera is mounted to the EE, is another example of camera to robot calibration problem to which conventional solutions that we explained earlier are also applicable. Partially similar to our system, a recent method works on hand-in-eye calibration via deep learning [4]. This method uses synthetic data and 2D images, and a different robot-camera setup.

## 3   Method

Our system consists of two main stages; (1) single frame EE 6-DoF pose estimation and (2) multi-frame extrinsic calibration. The output of the former along with forward kinematics is enough to get a calibration estimate but multiple frames increases robustness. Fig. 2 shows our single frame EE pose estimation and Fig. 3 shows our calibration workflows.

For EE pose estimation, we first segment the EE from a point cloud using a semantic segmentation approach. Then we utilize two approaches to estimate the EE pose. The first one does EE rotation prediction followed by a point cloud transformation step to get the EE translation. The second one extracts keypoints from the EE and matches these with reference points to calculate the EE pose. Both approaches are followed by an ICP step initialized with the EE pose estimates to match the EE points and the EE CAD model. These steps are summarized in Fig. 2.

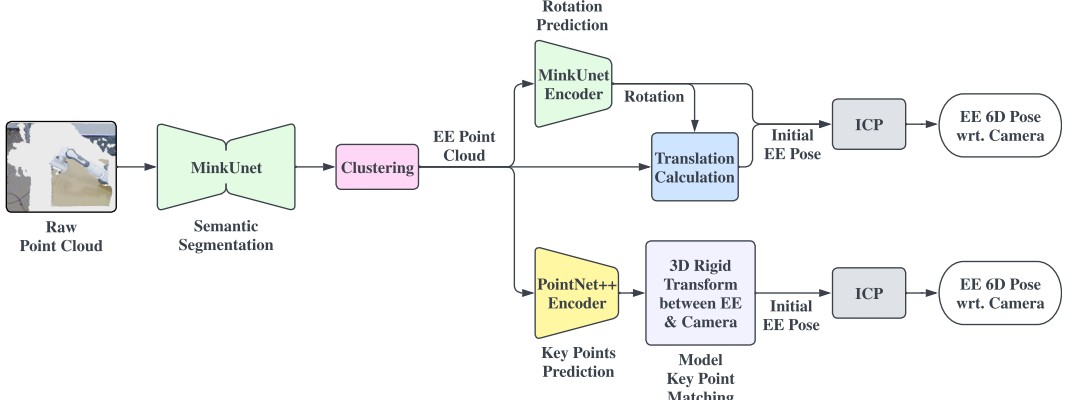

Figure 2: Single frame prediction architecture.

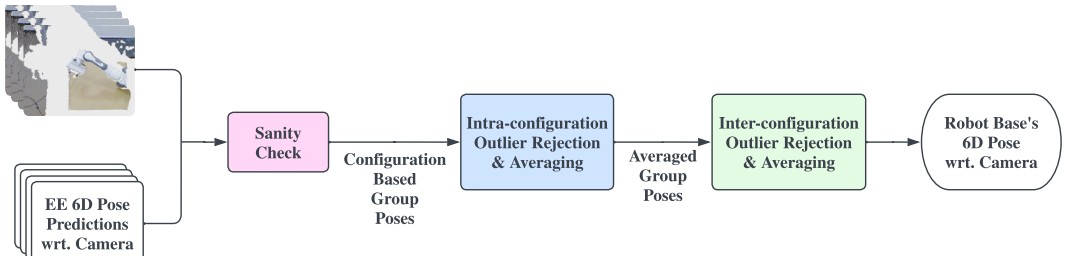

Figure 3: Calibration architecture.

For calibration, the system collects multiple frames and the corresponding EE pose estimates as described above. To increase robustness, sanity check and outlier detection steps are employed. The remaining poses are then averaged in the rigid-body transformation space (SO(3) for rotation and $\mathbb{R}^3$ for position), to get the final extrinsic calibration result. These steps are summarized in Fig. 3.

## 3.1 Data Collection

Our EE segmentation and pose estimation rely on learning and require robot specific data. We collect ground truth semantic segmentation labels, EE poses wrt. the depth camera and keypoint information. We collect this data from a real world setup to eliminate the need for simulation.

The first step is to perform an initial calibration. This can be done by existing approaches. We use a combination of markers, manual tuning on RViz [19] and ICP [20] towards this end. This is the only manual step that is needed and is only required for data collection. This is only done once per camera pose we want to collect data for. Once our system is up and running, no manual tuning is needed. This calibration information gives us the robot poses with respect to the camera frame.

The second step is to collect semantic segmentation data. We label points as background, EE and the rest of the robot automatically. Background points are obtained by utilizing frames without the robot. Then, the robot is moved to multiple visible positions to collect more frames. Semantic segmentation data for the robot arm is generated by subtracting the background points from every frame [21]. The EE points are further obtained by transforming the EE bounding box using the calibration information and taking the points inside this box[1].

The third step is to generate keypoint information. We extract a total of six keypoints; four are located at each corner of the EE and two are located at the tip of the gripper fingers as shown in Fig. 4. During training, we have access to the calibration. As such, we know the pose of the EE in the camera frame. We also know the reference keypoint locations with respect to the EE as they are

---

[1]When subtracting the background points is not possible, the CAD model of the robot, joint positions and the calibration information can be used to get the arm and EE points, and consequently background points.

fixed and use this information to get reference points in the camera frame. The EE points closest to these reference points are selected as keypoints, as long as their distances are below a threshold.

## 3.2 End-Effector Segmentation

The EE pose estimation starts by extracting the semantic segmentation information from an input point cloud. The outputs of this step are the EE, rest of the arm and the background labels for the points as shown in Fig. 1. For this step, we utilize a sparse version of the Unet architecture named the Minkowski Unet network [22, 23, 24], specifically the MinkUNet18D [22] architecture trained with the raw point clouds as the input and the collected semantic segmentation data as the target. We then apply linkage clustering [25] to points classified as EE in order to reject superfluous predictions.

## 3.3 Pose Estimation with Rotation Prediction and Transform Calculation: RPT

We first predict the EE rotation from EE points by utilizing the encoder part of another MinkUNet18D backbone with the same configuration. We train this with the segmented EE points as input and the EE rotation wrt. camera as target, using the loss of the PoseCNN model [15].

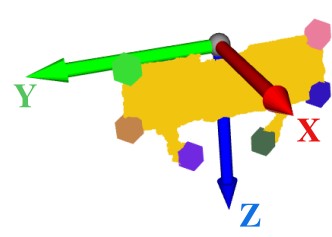

To calculate the EE translation, we use the output of the learned rotation prediction model and the geometric relationship between the EE frame and certain EE points on the bounding box, assuming that the EE is not occluded. We use the predicted rotation to rotate the EE points back to their non-rotated pose wrt. the camera. This is done by applying the Eq. 1 on each EE point, where $R^{EE}$ is the rotation prediction of the model, $P^{EE}$ is the translation of an EE point and the $\bar{P}^{EE}$ is the resulting rotated position. Both $P^{EE}$ and $\bar{P}^{EE}$ are in the camera frame.

Figure 4: End-effector segmentation (yellow), end-effector frame, and keypoints (colored hexagons).

$$\bar{P}^{EE} = (R^{EE})^{-1} \times P^{EE} \tag{1}$$

The EE points are along the surface of the EE. If we assume that the EE is fully visible, we can use the geometric relationship between the EE surface and the EE frame to get the EE translation. In the non-rotated pose wrt. the camera frame, the EE frame is 0.015m inside the surface along the $x$-axis, centered along the $y$-axis and closest to the camera along the $z$-axis (see Fig. 4). We use the following equations to capture this relationship where $\bar{t}^{EE}$ and $\bar{P}^{EE}$ denote the axes positions of the EE frame and EE points in the rotated frame, all in the camera frame.

$$\bar{t}_x^{EE} = \max\left(\bar{P}_x^{EE}\right) - 0.015 \tag{2}$$

$$\bar{t}_y^{EE} = \frac{\max\left(\bar{P}_y^{EE}\right) - \min\left(\bar{P}_y^{EE}\right)}{2} \tag{3}$$

$$\bar{t}_z^{EE} = \min\left(\bar{P}_z^{EE}\right) \tag{4}$$

We finally rotate $\bar{t}^{EE}$ back to get the EE translation, $t^{EE}$, wrt. the camera using Eq. 5.

$$t^{EE} = R^{EE} \times \bar{t}^{EE} \tag{5}$$

Calculating the EE translation with such an approach is efficient and accurate given that the rotation predictions are accurate and that the EE is visible. However, full EE visibility is not a given for all the frames and as such, this method cannot be trusted for single frame pose estimation by itself.

## 3.4 End-Effector Pose Estimation via Keypoint Matching: KPM

In this approach, we first predict keypoints among the EE points. We use a dense point cloud encoder, PointNet++ [26] trained with segmented EE points as input and the collected ground truth keypoints as target. We then use a version of the least-squares fitting [27] algorithm to find the rigid transformation between predicted keypoints and the reference points, which gives us the EE pose wrt. the camera. This is only possible when there are at least four high quality keypoint predictions.

### 3.5  Pose Estimation Refinement via the Iterative Closest Point Algorithm

We use the ICP algorithm [20, 28] to improve the pose estimation performance. This algorithm requires three inputs; the initial pose estimate (needs to be fairly accurate), a source point cloud, and a target point cloud. The initial pose estimate comes from either the RPT or the KPM methods, and the target point cloud comes from the EE segmentation. To generate the source points, we convert the CAD model of the robot's EE, provided by the manufacturer [29], to a point cloud, and transform the points based on the current EE configuration. The ICP algorithm outputs the transformation needed to match the source and target points which we use to refine the pose estimates.

### 3.6  Camera to Robot Base Calibration

A single EE pose with respect to the camera frame can be used to calculate the transformation between the camera and the robot using Eq. 6, where $T_X^Y$ represents the homogeneous transformation between frames $X$ and $Y$, and the letters $B$, $C$, and $EE$ correspond to the robot base, camera and the EE respectively. The EE in the base frame, $T_B^{EE}$, is obtained by forward kinematics and the EE in the camera frame, $T_C^{EE}$, is estimated by either the RPT or the KPM methods.

$$T_C^B = T_C^{EE}(T_B^{EE})^{-1} \tag{6}$$

However, a single frame is not robust enough to get a good calibration. We move the robot around and capture multiple frames at each robot pose to get more accurate estimates. We first run the semantic segmentation model and perform a sanity check on the output. If the number of EE points or the size of the EE bounding box are below their respective thresholds, we remove the frame. This is done to eliminate frames where the EE is not visible enough for accurate pose estimation.

After the sanity check, we group the predictions based on the robot configuration (we capture multiple frames per robot pose). We use a Z-score outlier detection algorithm using the absolute deviations about the median [30] for each translation axis of the camera to robot base predictions, per group. For rotation prediction outlier detection, we apply outlier detection on the rotational distances of every prediction to a reference unit quaternion. We remove predictions that are marked as outliers. Then, the remaining camera to base transformations are averaged by computing the arithmetic average for translation predictions and utilizing quaternion averaging [31] for rotation predictions. This gives us individual calibration estimations for each robot configuration.

Lastly, we employ the same outlier detection and averaging steps to the calibration estimates of each robot configuration to get the final estimate.

## 4  Evaluation

We evaluate the semantic segmentation, single pose estimation and camera calibration performances of our system when trained with three camera poses. We test our approach with ground truth segmentation labels as well to gauge the effects of the accuracy of the segmentation model on the pose and calibration estimation. We also provide results without the ICP step to highlight its benefits and compare to a AR-tag based classical baseline. Lastly, we perform evaluations with models trained on data from individual camera poses.

We use the translation error ($\epsilon_t$), rotation error ($\epsilon_R$) and the average distance (ADD) metrics to measure the pose estimation and calibration performances. The $\epsilon_t$ is the Euclidean distance between the ground truth translation and the predicted translation. The $\epsilon_R$ is the minimum rotation between the ground truth rotation and the predicted rotation. ADD is the average of point-to-point Euclidean distances between the EE points transformed with the ground truth pose and the predicted pose [32]. To measure the accuracy of our semantic segmentation model, we use accuracy, precision and recall.

### 4.1  Collected Data

Our hardware setup only includes a Kinect V1 and a Franka Emika Panda robot arm as shown in Fig. 1. We collect around 7000 frames from 3 different camera poses. The cameras are placed on three sides of the robot setup, as shown in Fig. 5. We reserve 1000 frames for validation to control overfitting. To increase generalization, we apply 3D data augmentation methods such as elastic distortion, noise injection and point dropouts during training. We also collect test data from

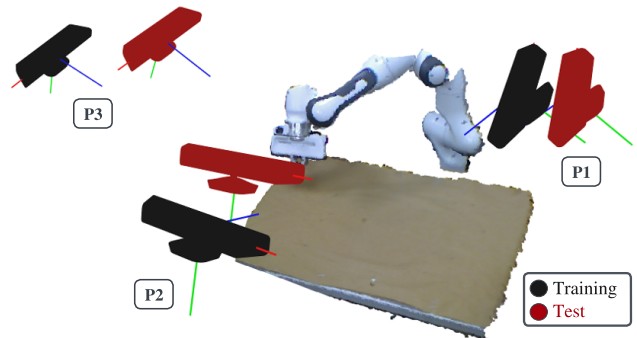

Figure 5: Visualization of Kinect camera positions during training and test data collection.

| | P1 Test | | P2 Test | | P3 Test | |
|---|---|---|---|---|---|---|
| | $\Delta_t$ [cm] | $\Delta_R$ [°] | $\Delta_t$ [cm] | $\Delta_R$ [°] | $\Delta_t$ [cm] | $\Delta_R$ [°] |
| P1 Training | 5.15 | 14 | 95.12 | 71.33 | 173.49 | 119.07 |
| P2 Training | 102.7 | 96.32 | 5.88 | 18.31 | 88.12 | 95.37 |
| P3 Training | 175.35 | 170.16 | 86.44 | 112.66 | 6.2 | 42.5 |

Table 1: Relative translation and rotation differences between camera pose pairs.

additional 3 cameras poses (see Fig. 5), 6 locations per pose and 10 frames per location for a total of 180 frames. The translation and rotation differences between training and test camera poses are given in Tab. 1. We additionally collect test data using an ArUco tag attached to the EE in the same test poses. We chose a relatively large marker to be robust to camera distance as shown in Fig. 6.

The training data is collected with an initial calibration. As we do not have access to high-quality markers (e.g. a motion capture system), this may suffer from all the issues laid out in this paper. The ADD of the training data is calculated as $0.75cm$, which implies a relatively high quality data set.

| Class | Precision | Recall | Accuracy |
|---|---|---|---|
| EE | 0.96 | 0.99 | 1.00 |
| Arm | 0.87 | 0.99 | 0.99 |
| BG | 1.00 | 0.99 | 0.99 |
| All | 0.94 | 0.99 | 0.99 |

Table 2: Semantic segmentation res.

Semantic segmentation performance is important as it is the first step and its EE segmentation output is used as input to the following steps. We use precision, recall and accuracy to assess the performance of segmentation. Tab. 2 presents these results. Overall, the segmentation model performs well. The recall ratio for EE segmentation is $\sim 100\%$, i.e., the model is able to catch all the EE points.

The pose estimation and extrinsic calibration performances are calculated with both the model and ground truth segmentation labels (Table 3). Tab. 4 shows that there is around $0.05cm$ and $0.5°$ differences between the average errors. These are negligible when the camera noise is taken into account [12]. Combining this with the segmentation performance, we conclude that the segmentation model performs well enough to generate reliable inputs for the subsequent steps.

## 4.2 Single Frame Pose Prediction and Multi-Frame Calibration

Tab. 3 shows the pose estimation performances of the RPT and KPM approaches with and without the ICP post-processing step and with ground truth and predicted semantic segmentation labels.

**ICP step provides considerable improvement for pose estimation:** The addition of the ICP step improves both algorithms in all metrics in both segmentation input cases. In addition, the absolute performance is very impressive with about $1cm$ translation and less than $3°$ rotation errors.

| | $\epsilon_t$ [cm] | | $\epsilon_R$ [°] | | ADD [cm] | |
|---|---|---|---|---|---|---|
| Method | GT | Model | GT | Model | GT | Model |
| RPT | 2.36±0.36 | 2.49±0.77 | 7.75±5.05 | 7.18±4.62 | 2.47±0.52 | 2.58±0.90 |
| RPT+ICP | 0.87±0.20 | 1.01±0.32 | 3.20±2.42 | 3.47±2.75 | 0.97±0.30 | 1.07±0.35 |
| KPM | 1.35±0.37 | 1.43±0.48 | 7.46±1.54 | 7.44±1.44 | 1.53±0.33 | 1.60±0.47 |
| KPM+ICP | 0.96±0.27 | **1.00±0.27** | 2.42±0.98 | **2.74±1.59** | 0.99±0.26 | **1.04±0.27** |

Table 3: EE pose estimation results with ground truth (GT) and predicted (Model) semantic labels.

| | $\epsilon_t$ [cm] | $\epsilon_R$ [°] |
|---|---|---|
| AR-Tag | 1.83 | 3.26 |
| Ours | 2.35 | 3.37 |
| Ours + ICP | 0.74 | 1.69 |
| GT + ICP | 0.77 | 1.19 |

Table 4: Calibration results.

**High Performance Multi-Frame Calibration:** The calibration results are given in Tab. 4. The average translation and rotation error between the predicted calibration and the ground truth calibration are $\epsilon_t = 0.74cm$ and $\epsilon_R = 1.69°$, respectively. As expected, the ICP step significantly improves performance, and semantic segmentation and ground truth inputs perform similarly. This absolute performance would allow for most manipulation applications outside of assembly or high precision tasks. These results are affected by the sensor noise, and potentially the intrinsic calibration and the initial extrinsic calibration errors. We argue that a less noisy camera and/or an active marker system for initial calibration is needed to go beyond these values.

We also compare our system with an AR-tag based baseline. The AR-tag results in Tab. 4 are with outlier detection but without the ICP step. These results show that the AR-tag baseline holds $0.5cm$ advantage in translation and is on par in rotation compared to our base method. We chose a relatively large marker to be robust to distance and large orientation changes. This alters the EE shape considerably (see Fig. 6), and makes ICP infeasible. A smaller tag degrades the performance considerably with distance to camera. If the ICP algorithm was feasible with the large marker, the final calibration results would be on-par. Thus, our method can achieve similar results to a classical marker based method while being easier to use and requiring much less manual effort down the line (a large

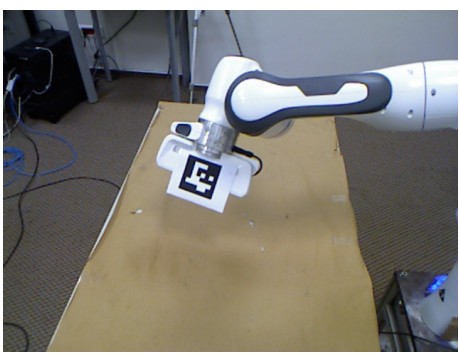

Figure 6: EE with the AR-tag attached

marker needs to be removed for work and re-attached for calibration). Our system would have better performance than an AR-tag based approach with a smaller marker that is more practical to use.

**Discussion Pertaining to Existing Work:** It is difficult to conduct an apples-to-apples comparison between our work and the existing learning-based methods since the setups are significantly different. This is due to simulation requirements, lack of point cloud data in existing datasets and the difference in predicted outputs (e.g. full arm vs EE). However, a qualitative discussion can be made over the common ADD object pose estimation metric.

RPT and KPM approaches outperform state-of-the-art object pose detection methods [15, 16]. The performance gets much better when the ICP step is included. We also outperform a similar single frame pose estimation method [2]. With an ADD threshold of $2.0cm$, we get $100\%$ pose estimation accuracy whereas they get less than $40\%$ for Kinect V1. However, the generic object pose detection methods deal with multiple objects [33] and Lee et al. [2]'s result is for the entire robot arm, whereas we are only looking at the EE.

### 4.3   Training with Data from Individual Camera Poses

The results we presented so far were obtained from data collected from each training camera location (see Fig. 5). These locations cover either side and the front of the robot. They also cover a range of distances to robot base where P1 is closest and P3 is the farthest. However, this raises the question of calibration performance without such coverage.

To answer this, we train different models using data from each training camera location and test it on all the test data (e.g. train with P2-training, test on P1-test, P2-test and P3-test). As given in Table. 1, there are significant translation and rotation differences between the poses with the only similarity being the pitch axis. Tab. 5 shows the calibration results with the full system.

The results show that, the closer the train-test locations, the better the performance with an exception for P1-train and P1-test. The reason is that the P1-train pose is very close to the setup, and as such, its training set lacks data from farther EE poses. System trained with P2 data achieves the best overall results, as expected, since it is in the middle both translation-wise and rotation-wise. Furthermore, its performance is not too far from the system trained with data from three camera poses.

The results imply that, a careful selection of the training camera pose is important but a single camera is enough with a slight sacrifice in performance while reducing the initial calibration effort.

| | P1 Test | | P2 Test | | P3 Test | | Average | |
|---|---|---|---|---|---|---|---|---|
| | $\epsilon_t$ [cm] | $\epsilon_R$ [°] | $\epsilon_t$ [cm] | $\epsilon_R$ [°] | $\epsilon_t$ [cm] | $\epsilon_R$ [°] | $\epsilon_t$ [cm] | $\epsilon_R$ [°] |
| P1 Training | 1.32 | 0.64 | 1.64 | 2.81 | 4.21 | 5.63 | 2.39 | 3.03 |
| P2 Training | 0.84 | 0.60 | 0.70 | 1.82 | 1.61 | 2.32 | 1.05 | 1.58 |
| P3 Training | 1.82 | 1.20 | 2.28 | 2.12 | 0.93 | 1.89 | 1.68 | 1.74 |

Table 5: Calibration prediction results for respective training and test sets.

# 5 Limitations and Future Work

The quality of the training data depends on the initial calibration, RGBD sensor noise and forward kinematics error (e.g. due to gear backlash, joint sensor calibration, etc.). A careful calibration is needed for any robotics task so there is no extra manual effort to use our system. However, this should be performed carefully. We used a relatively noisy RGBD camera [12]. We expect a better camera to yield better results. The user should be aware of the forward kinematics errors and, if they are severe, should take this into consideration. In such cases, training data should be processed with ICP and only single EE frame pose estimations should be used. This affects any type of calibration.

Our system requires that the EE is not occluded during calibration. The RPT method would not be able to calculate accurate translations otherwise. The KPM method is more robust against occlusions as it can work with four keypoints. If occlusions are expected, more keypoints should be selected and RPT should be disabled.

We collect data with the robot arm in different joint configurations for the calibration operation. These joint configurations should be diverse for a better calibration estimate. The test set in this work is collected from manually selected joint configurations. A fully automated system should choose these by itself while keeping the EE in the camera frame.

Our system assumes that there are no objects in the camera view during calibration since we do not collect object data. We use a relatively high-capacity model, MinkUnet18D [22], for the semantic segmentation task involving only three classes. As such, it is prone to overfitting. An out of sample input (i.e. a point cloud with objects) would be difficult to handle. Using lower capacity models such as MinkUNet101 and MinkUNet14A deteriorated our semantic segmentation performance. Superimposing object point clouds as a data augmentation approach is possible in to circumvent this limitation. However, this is a mild assumption to handle extrinsic calibration prior to any application.

Currently, we provide the desired keypoints by hand. Even though it is enough to do this once for each EE, it is not the most ideal approach since the user may not be familiar enough with robotics to do so or the selected keypoints may not be easily distinguishable. An automated keypoint selection algorithm can be added in the future to remove this burden from the user.

We are not combining the outputs of the RPT and KPM methods to get a single frame EE pose estimation and only average them during calibration. A future work is to investigate the potential of using them together, for example to have them share a backbone, feed the output of the rotation network into the KP network, etc.

# 6 Conclusion

We presented a learning-based extrinsic calibration system that does not require fiducial markers, additional hardware or simulation. Our system collects and labels its own data. The only manual steps are initial calibration for data collection and keypoint selection. Our system performs extrinsic calibration from multiple frames and employs steps to increase its robustness. In addition to extrinsic calibration, our system outputs high quality arm and EE segmentation information and camera-to-EE pose estimation. The extrinsic calibration challenge is never-ending, especially for low cost or multi-user setups and with our system, the user needs to manually calibrate only once.

We tested our approach in different camera configurations then the ones used for training. Our results showed that the ICP algorithm significantly improves estimation performance and the absolute calibration results are close to the limit of what is possible with the used depth camera. We also showed that a carefully selected single camera training pose is enough. The main limitations of our approach are the lack of object consideration, which is mild if the only aim is extrinsic calibration, and manual keypoint selection.

**Acknowledgments**

This work was supported by KUIS AI Center computational resources. The authors would also like to thank Onur Berk Töre and Farzin Negahbani for their infrastructure support and work on an earlier version of the system.

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
