# OpenReview forum: "Learning Markerless Robot-Depth Camera Calibration and End-Effector Pose Estimation"
_robot-learning.org/CoRL/2022/Conference — CoRL 2022 Poster_

### Official Review · Reviewer_HmKT · 2022-07-20

**Originality:** Good
**Technical Quality:** Good
**Clarity Of Presentation:** Good
**Impact:** 3

**Recommendation:**

Weak Accept: I recommend accepting the paper, but will not argue for my recommendation if the majority of other reviewers have a different opinion.

**Summary:**

The authors are tackling the problem of calibrating an external camera looking at a robot arm. They use learning based methods along with geometric considerations to perform the calibration.  Their method allows calibration to be performed quickly, without the need of specific calibration objects such as calibration tags.


**Issues:**

It would be great if the authors could address the following:

- Line 112 : is that alternative used? Or is simply a proposed alternative in case the previous steps cannot be performed. This could be made clearer.
-It seems only point clouds are used throughout the paper as input modalities, this should be stated explicitly somewhere.

- In line 130 I’m not sure that custom is the correct word? Perhaps modified? If not then custom to what? This paper?

- Why is the accuracy not done per class in Table 1?

- How prone is the errors in this method to the errors in the initial labelling? Has there been any evaluation of the calibration  performed for the labels? This also seems to be an issue with the evaluation of the method : Is the reported  accuracy relative to that label calibration?

- What frames of reference are used in needs to be made a bit clearer throughout the paper. For example, in line 136, is the rotation in the base frame or the camera frame? It is the same for the equations, I suggest adding the frame explicitly in the notations of the variables (while explaining the notation), so that everything is clearer.

- Lines 143-148 are not entirely clear, what is the canonical location? “The centre of the EE frame”  is not clear, a frame is simply a point with three axis, how can it have a center?  Similarly, how  are the rules  lines 153-154 obtained?


**Quality Of The Limitations Section:**

Additional details required

**Reviewer Expertise:**

4: The reviewer is confident but not absolutely certain that the evaluation is correct

**Robotics Focus:**

Highly relevant to robotics but no hardware experiments

**Strengths And Weaknesses:**

**Summary of Strengths**
- The paper tackles an interesting problem that can have a large impact in robotics.
- An interesting and informative method is presented, combining learning components and geometric considerations.

**Summary of Weaknesses**
- It would have been better to do an actual comparison/benchmarking against papers performing the same task, namely Labbe et al, and Lee et al.
- The evaluation results seem to be dependent on the quality of the “ground truth” predictions, which can also be prone to errors, and there is no provided estimate of how large these errors may be.
- Although one can follow what is being done, some points on the presentation can be improved.


**Comments**

- The extrinsic calibration and EE pose estimation are very closely related (by equation 6) , yet they are contrasted in the introduction without further explanation: This is confusing when reading it the first time and can be made clearer. Similarly, having the pose estimation and the extrinsic calibration in the same table later on compounds on the confusion. What is especially unclear is which rows of the table are used to perform the calibration. From  my understanding, multiple estimates of the ICP variants of both pose estimation heads are used independently to obtain the calibration through averaging and outlier rejection, but this is not very clear overall.

- Both the method and evaluation seem to be dependent on a “ground truth labelling” but we have no information on the accuracy of those ground truth values. In the experiment section in lines 214-216 it is unclear what the “ground truths” are: Is it the EE pose obtained through the robot kinematics? Is the calibration label obtained as described in line 101? This would need to be made clearer to better understand the results. The authors recognise this issue in lines 266- 269, but I believe this point needs to be expanded on and talked about earlier on, and it should also be added in the limitations of the paper.

- Lines 247 - 249 : It would be good to provide some insight as to why this may be the case, either experimentally, or by conjecture.




**Summary Of Recommendation:**

The method presented by the paper is interesting and informative, and the problem tackled is important. However, there are some important points that are currently unclear in the paper, such as the possible reliance of the presented results to labelling that may also be error prone, without an estimate of what that error may be.

---

> ### Author Response · Authors · 2022-08-18
> **Initial Rebuttal and Action Plan Related to Reviewer Comments**
>
> We thank the reviewer for their invaluable time and useful feedback.
>
> Overall: The main concern of the reviewer is (1) clarity, (2) quality of GT data and (3) lack of benchmarking. We are going to streamline our paper with all the reviewers' comments for (1). We argue that this level of calibration information is already used in labs but will also include an evaluation on the training data for (2). For (3), we argue that methods are not directly comparable due to data sources, setup and the need for calibration but we will try to add a comparison using an existing model.
>
> Weakness Point 1:
> Unfortunately, there is not a direct comparable method since they use RGB data and require simulation. The setup needed to compare the results would require significant effort. The datasets they use also do not include depth information. However, Lee et al.'s learned model may be used with our data for testing purposes as they use the same robot. We are investigating this and will present the results to the reviewers if feasible. Their method predicts joint positions to calculate pose. Since the robot occludes itself in our setup, it is difficult to steel-man this method.
>
> Weakness Point 2:
> We are manually tuning the initial calibration and perform ICP on top of it. To the best of our knowledge, getting high precision calibration data requires expensive setups which we (and many robotic labs) do not have access to. However, most applications outside factory floors, can work with simpler and lower cost yet less precise methods. AR-tag or checkerboard pattern based methods are common examples. As such, the GT we use is already within acceptable precision range. Using ICP further increases this precision considerably.
>
> However, as the reviewer suggests, it is important to have a perspective on how good this GT is. As such, we are going to calculate the ADD metric for each training point and present the results soon. We are open to other suggestions as well.
>
> Weakness Point 3:
> We are going to address this issue by clarifying certain parts and by providing better figures. We will share the new figures.
>
> Comments Point 1:
> We did not intend to contrast the two, they are related. The reviewer's understanding is correct about calibration. We will clarify both points.
>
> Comments Point 2:
> See response under weakness point 2. We will add both a relevant evaluation on the goodness of the GT and discussion about its limitations.
>
> Comments Point 3:
> This is a very good point. The translation calculation of RPT inherently assumes that the end-effector mostly visible which may not be the case. This is the most likely reason why KPM is better than RPT. On the other hand, RPT directly optimizes a rotation loss which is the most likely reason why RPT is better than KPM for this. We will add this discussion in the paper
>
> Issue Point 1:
> We did not use the alternative. We will clarify this in the text.
>
> Issue Point 2:
> We agree that the usage of the word "custom" is wrong and will fix it.
>
> Issue Point 3:
> We will add accuracy as "one-vs-all"
>
> Issue Point 4:
> See our response under weakness point 2 and comments point 2. In short, we will provide an ADD based analysis for this.
>
> Issue Point 5:
> Thank you for pointing this out. Everything is in the camera frame and we will clarify this in the text.
>
> Issue Point 6:
> We agree that the discussion is confusing. By canonical pose we actually mean "non-rotated" pose. For the centre comment, we should have written "Then we use the relationship between the EE frame and the points ... ".
>
> The rules are calculated using the CAD model. For a non-rotated model, we know the offsets (i.e. the geometric relations) of its bounding box limits to the EE frame. When we get a rotation prediction, we rotate the end-effector points (obtained via semantic segmentation) back, so that they correspond to the EE pose with no rotation. We then use the geometric relation between the EE frame and the relevant points to get the translation
>
> In order for this to work, we are making an assumption that most of the end-effector is visible in the frame during calibration. We handle the cases where this is not the case with filtering/pre-processing. Note that it is not difficult to extend this to other robot if there is a CAD model available.

---

> ### Author Response · Authors · 2022-08-28
> **Final updates on our work**
>
> We shared our modified paper and the details as a comment to meta review. Please use https://openreview.net/forum?id=eI8CZ2s267o&noteId=iAa3zy_4SXN to view the modified paper and our comments.

---

### Official Review · Reviewer_LhSL · 2022-07-27

**Originality:** Good
**Technical Quality:** Good
**Clarity Of Presentation:** Good
**Impact:** 2

**Recommendation:**

Weak Accept: I recommend accepting the paper, but will not argue for my recommendation if the majority of other reviewers have a different opinion.

**Summary:**

The paper tries to address the issue of Robot-to-Camera calibration for a Robot manipulator to a third-person Depth Camera using point cloud-based segmentation of the robot end-effector (EE). Initially, robot-specific data is collected to train the segmentation models to segment and predict specific key points on the robot EE. Using these outputs, additional networks are used to predict the transformation from the EE to the camera, which is refined using ICP with a CAD model of the EE. Using this and the forward kinematics of the robot, the transformation from the robot base to the camera is derived. Finally outlier rejection and averaging is used for multiple frames and robot poses to get the final calibration. The approach shows good performance (within cm range for translation and 3 degrees for rotation calibration) on a Franka robot arm.

**Issues:**

- The description of the data collection on how the key points for the EE are obtained was not clear to me. Is it a 3D bounding box fitted over the points from which fixed points relative to the bounding box frame are used?
- The translation prediction seems quite ad-hoc. Were these values something that was chosen based on the ideal locations of the EE CAD model points or something like that? The paper would benefit from a description of this process and a also few lines on how this could be extended to other robotic EEs, which can help make the approach more generalizable.
- The approach could be made more principled by maybe using the outputs of the Key Point Matching (KPM) in some form to possibly improve the prediction of the Rotation Prediction and Transform Calculation (RPT) since I would imagine that using particular key points of interest could help the RPT to predict more accurate poses. Or potentially using an estimate of the transformation to improve the key point detection for further refinement of the predicted transformation.
- From the code, it seems like the ICP is not used for training the networks. I would imagine that the networks could benefit by using the output after ICP for training.

**Quality Of The Limitations Section:**

Limitations are addressed clearly

**Reviewer Expertise:**

4: The reviewer is confident but not absolutely certain that the evaluation is correct

**Robotics Focus:**

Sufficient demonstration on hardware

**Strengths And Weaknesses:**

### Strengths
- Present a good pipeline that can be trained end-to-end for robot-to-camera calibration. Once trained for a given robot and camera, the approach can, in theory, be applied as and when needed for different experimental settings without the need for re-calibration each time.
- Overall good performance:
    * Shows good segmentation between end-effector, robot and background points.
    * Achieves low calibration error on their specific setting.
    * Works well even with some occlusions (as seen in the video).
- Limitations are considered and well addressed at the end of the paper, with respect to overfitting, occlusions etc.
- Overall writing and organisation of the paper are fairly well done and easy to follow.
- **The source code is also provided**

(Post-Rebuttal Update)
- The comparison with the AR Tags shows good performance especially after using ICP which is similar to using Ground Truth segmentation with ICP.
- The application scenario is better highlighted in Section 4.4 where the approach is trained from one view point and tested on other view points showing the relative generalization capabilities.

### Weaknesses
- ~~The current approach needs extensive training data (10k robot frames) along with the initial ground truth generated for each robot. This is done using Markers, ICP and Manual tuning.~~
- ~~There is no comparison with existing works, or even with some baselines like plain ICP, (AR) markers or a combination of these two or a  naive segmentation+ICP, etc. If the approaches are used to just initialize ICP, how much of an advantage does this give compared to say manually or semi-autonomously initializing it? This makes it difficult to gauge the effectiveness of the proposed approach.~~
- ~~The paper proposes two approaches for predicting the transformations, one using segmented EE points from the point cloud and the other by using the predicted key points of the EE. However, I feel that the paper would benefit from a more principled coupling of the two approaches since they are interdependent, for example, either by jointly training them for a unified pose estimate or by using the segmentation-based prediction to initialise the key point-based prediction which is then subsequently refined. Currently, they are just used to generate pose estimates of multiple frames which are then fed to an outlier rejection and post-processing. While this process works, exploring these kinds of additional approaches for improving the pose prediction could make the approach more principled and novel.~~

(Post-rebuttal Update)
The above weaknesses were taken care of in some form (either by experiments or by including them in the limitations section).


**Summary Of Recommendation:**

The paper solves a relatively important problem used in the pipeline of robotic experiments. Having gone through such struggles myself in setting up and properly calibrating a robot w.r.t. an RGBD camera, I understand the relevance of the problem and the impressiveness of the results. ~However, the lack of evaluation with baseline methods (like the ones mentioned above) and the extensive requirement of calibrated data do not fully convince me to argue for a strong acceptance. These are important in order to gauge the effectiveness of the method in making it a fast reliable approach for calibration.~

(Post-Rebuttal Update)
Some of the concerns have been addressed and the comparison and the accuracy on individual camera poses shows the effectiveness of the proposed approach.

---

> ### Author Response · Authors · 2022-08-18
> **Initial Rebuttal and Action Plan Related to Reviewer Comments**
>
> We thank the reviewer for their invaluable time, useful feedback and important insights.
>
> Overall: The main issues that the reviewer has with the paper is the (1) lack of a comparison against a baseline and (2) requirement for calibrated data. For (1), we are taking the necessary steps and will share our results soon. For (2), we argue that it is the price of using learning and removing the need for simulation data.
>
> Weakness Point 1:
> Data collection is unfortunately the price to pay for learning and removing the dependency on the simulation altogether. However, this data collection step is mostly automatic. As the reviewer noted, roboticists already need to perform extrinsic calibration which is the only manual step we require. As such, data collection does not require significant additional effort. We argue that this investment would save the user a lot of time down-the line.
>
> Weakness Point 2:
> - For classical baseline: Most classical approaches are marker based and we argue that the use cases are different. We would need to keep the marker permanently positioned on the robot to make them equivalent and would also need to calibrate the marker position with respect to the robot. Existing learning-based methods do not also include comparison against such a classical approach. We think that this is due to the aforementioned issues.
>
> However, we are looking into ways to incorporate an off-the-shelf AR-tag based approach to use as a baseline while steel-manning this baseline as much as possible. We believe this is doable before the rebuttal's end will share the results accordingly. ICP doesn't work well with random initialization but it does with an AR-based initialization which is what we will do.
>
> - For learning-based baselines: Unfortunately, there is not a direct comparable method since they use RGB data and require simulation. The datasets they use do not include depth information. However, Lee et al.'s method may be used with our data. We are investigating this and will present the results to the reviewers if feasible. Their method predicts joint positions to calculate pose. Since the robot occludes itself in our setup,  it is difficult to steel-man this method.
>
> Weakness Point 3:
> This is a very valid point. Due to the limited time for rebuttals, we believe that the idea proposed by the reviewer would be better suited as a next step for our work. We can add this as a discussion both in the limitations part and the future work part. Some of the challenges are as follows:
> - Since the models do not share a backbone, it is not straightforward to train them together. We picked the encoders/backbones based on published performances. A common backbone may be investigated.
> -  Going from a single pose estimate is difficult as we would need a differentiable approach to do this end-to-end.
> - Feeding the orientation output or the pose estimation output as an input to the keypoint detection model is an interesting idea which warrants exploration. This would entail architectural changes and re-training which is doable. However, we are prioritizing more detailed analysis and  baseline comparison.
>
> Issue Point 1:
> We agree that our description is not clear. We will clarify it.
>
> During training, we have access to calibration. As such, we know the pose of the end-effector in the camera frame. We also know the keypoint poses with respect to the end-effector as they are fixed. We can therefore get the keypoints, using these transformations, in the camera frame.
>
> Issue Point 2:
> We will clarify this in the paper. The values are obtained from the CAD model. For a non-rotated model, we know the offsets of its bounding box limits to the EE frame. we are making an assumption that most of the end-effector is visible in the frame during calibration. We handle the cases where this is not the case with filtering/pre-processing. Note that it is not difficult to extend this to other robot if there is a CAD model available.
>
> We agree with the reviewers assessment of the benefits of further discussion and will include it in our paper.
>
> Issue Point 3:
> This is a great point and warrants exploration. However, due to time limits, we may not be able to incorporate it before the rebuttal's end. See also our response under Weakness Point 3.
>
> Issue Point 4:
> This is a great idea. We used the ICP to get the initial calibration and made sure that the camera or the robot did not move during data collection. However, this idea would compensate for any joint angle errors (i.e. errors in forward-kinematics). It can be argued that the error in joint readings are negligible compared to sensor noise but this needs further exploration as well. We argue that baseline and more detailed evaluations are more important but are open to suggestions by the reviewer

---

> > ### Comment · Reviewer_LhSL · 2022-08-25
> > **Comment on Classical baselines**
> >
> > Dear Authors,
> > Thanks a lot for the clarifications. When you say that "We would need to keep the marker permanently positioned on the robot to make them equivalent and would also need to calibrate the marker position with respect to the robot." would this not also be the same argument for using markers for the initial calibration during the data collection?

---

> > > ### Author Response · Authors · 2022-08-25
> > > **About classical baselines**
> > >
> > > Dear Reviewer,
> > >
> > > You are absolutely correct that the initial calibration with a marker also has the problem of figuring out the transformation between the marker frame and the end-effector frame. The usual practice is to get an approximate external calibration matrix and to improve this via manual tuning and the ICP algorithm. This requires some additional manual effort. In our approach, this is done once to collect the data.

---

> ### Author Response · Authors · 2022-08-28
> **Final updates on our work**
>
> We shared our modified paper and the details as a comment to meta review. Please use https://openreview.net/forum?id=eI8CZ2s267o&noteId=iAa3zy_4SXN to view the modified paper and our comments.

---

### Official Review · Reviewer_5rEJ · 2022-07-30

**Originality:** Fair
**Technical Quality:** Fair
**Clarity Of Presentation:** Good
**Impact:** 3

**Recommendation:**

Weak Reject: I recommend rejecting the paper, but will not argue for my recommendation if the majority of other reviewers have a different opinion.

**Summary:**

The paper presents a markerless method to perform hand-eye calibration through learning and estimating the end-effector pose. The idea is not new, which was previously researched in literature but has not applied to the extent presented in this paper. The main contribution is the realization of hand-eye calibration through direct learning of the pose of physical parts through CAD models and depth sensing. The paper is well-organized with a direct result and explanation, although the thoroughness of the results needs further explanation to consolidate the contribution of learning in such scenarios.

**Issues:**

- Revision in Paragraph 2, if “using markers is not an ideal solution for most applications,” then what solution is mainly used for robot hand-eye calibration? The logic here is misleading and confusing.
- Further analysis of the proposed method applied with cameras of higher accuracy as the baseline of comparison.
- Further analysis of calibration results compared with classical marker/checkboard-based method.
- Further analysis of the proposed method in dealing with occlusion and if learning can be used to resolve the problem in ways that classical methods cannot.
- What if simulation data is used, and what would the result look like using the proposed method?


**Quality Of The Limitations Section:**

Limitations are not well addressed

**Reviewer Expertise:**

5: The reviewer is absolutely certain that the evaluation is correct and very familiar with the relevant literature

**Robotics Focus:**

Sufficient demonstration on hardware

**Strengths And Weaknesses:**

Strength:
- Using the proposed method, if the robot’s CAD models are known (which is the case for most robots), no additional markers are needed to perform the calibration.

Weakness:
- Lack the in-depth analysis of the results is the main problem. For calibration, accuracy is the key here. For the proposed method, the system error is limited by the pose estimation of certain parts of the robot body, especially the end-effector. However, since only a Kinect V1 is used, it is difficult to evaluate the usefulness of the calibration accuracy. Does the error mainly caused by the sensor performance or caused by the algorithm’s performance in pose estimation? The estimated pose is also influenced by the sensor's data quality.
- Further comparison of the calibrated results against the typical/traditional calibration method using markers is not well presented. Considering this is a reasonably classical problem, this part of the comparison seems to be critically essential to justify the contribution from learning, which is not well-argued based on the current content.
- Further implementation of the method when applied to other robots. Although this is not that important for the paper's sake, it would be essential to see robot-specific characteristics in terms of design and occlusion, and how to make the process autonomous without interference by human operators.


**Summary Of Recommendation:**

The major problem is the thoroughness of the results, which is not well presented in depth to demonstrate or prove the contribution of learning in performing this calibration task. It is unclear if learning contributes significantly to the calibration accuracy or if superior calibration accuracy can be achieved using the proposed method even using a consumer-grade camera released years ago.

---

> ### Author Response · Authors · 2022-08-18
> **Initial Rebuttal and Action Plan Related to Reviewer Comments**
>
> We thank the reviewer for their invaluable time and useful feedback.
>
> Overall:
> The main issues that the reviewer has with the paper is the (1) lack of thorough results and (2) lack of a better RGBD sensor. For (1), we are going to present our detailed results soon. For (2), using a noisy sensor is to our disadvantage but we argue that our results would only get better with a better sensor. We may be able to evaluate with another sensor if the time permits but think that our results are quiet good already in terms of automatic calibration performance.
>
> Weakness Point 1:
> We are in the process of performing more detailed evaluations and will soon upload the new results accordingly.
>
> The Kinect noise issue is very real and we agree completely. We are in the process of trying to borrow another RGBD camera to test our methods. We are not sure if it can be ready by the rebuttal’s end. However, using a worse sensor is actually to our disadvantage. Our results can only increase with a better sensor.
>
> There is existing work investigating the noise of several RGBD cameras. For example "Comparison of Kinect V1 and V2 Depth Images in Terms of Accuracy and Precision" by Wasenmüller and Stricker published in an ACCV workshop published in 2016. We can add a discussion citing such work.
>
> Weakness Point 2:
> Most classical approaches are marker based and we argue that the use cases are different. We would need to keep the marker permanently positioned on the robot to make them equivalent and would also need to calibrate the marker position with respect to the robot. Existing learning-based methods do not also include comparison against such a classical approach. We think that this is due to the aforementioned issues.
>
> However, we are looking into ways to incorporate an off-the-shelf AR-tag based approach to use as a baseline while steel-manning this baseline as much as possible. We believe this is doable before the rebuttal's end will share the results accordingly.
>
> Weakness Point 3:
> This is a good point. We can overcome this with some initial human effort where we ask for a few canonical poses where the end-effector is fully visible and calculate the filtering/pre-processing parameters from these. Since some human effort is inevitable due to getting initial calibrations, we argue that this addition would be negligible.
>
> Issues Point 1:
> We will clarify this point to emphasize the repetitive nature of the calibration challenge. Furthermore, there is a learning-based approach for eye-in-hand calibration from last year's CoRL which we are citing. We can further add the relevant discussion.
>
> Issues Point 2:
> See the reply to weakness point 1
>
> Issue Point 3:
> See the reply to weakness point 2
>
> Issue Point 4:
> We are not claiming to obtain pose estimates when there is occlusion. We are handling these cases with pre-processing/filtering steps. One significant issue with occlusions is that ICP algorithm breaks down considerably. Occlusions are not rare but we argue that they can be avoided to get good calibration results prior to using the calibration for downstream tasks. We can add a discussion about this in the limitations part of the paper.
>
> Issue Point 5:
> This is an interesting question. One of our aims was to remove the dependence on simulation altogether since the noise models for RGBD cameras in simulation are usually not realistic, in addition to having good simulation environment and models. It can be argued that simulation data would reduce the amount of data needed from the real robot. We can add a discussion about simulation data to the paper.

---

> ### Author Response · Authors · 2022-08-28
> **Final updates on our work**
>
> We shared our modified paper and the details as a comment to meta review. Please use https://openreview.net/forum?id=eI8CZ2s267o&noteId=iAa3zy_4SXN to view the modified paper and our comments.

---

### Official Review · Reviewer_2Nyt · 2022-08-02

**Originality:** Fair
**Technical Quality:** Fair
**Clarity Of Presentation:** Poor
**Impact:** 2

**Recommendation:**

Weak Reject: I recommend rejecting the paper, but will not argue for my recommendation if the majority of other reviewers have a different opinion.

**Summary:**

This paper presents a learning-based system for calibrating the extrinsic pose of a robot arm and its end effector by using a depth sensor. The approach collects depth segmentation labels in order for it to identify the robot arm and end effector, and uses these to estimate the end effectors pose. This pose estimate is used in combination with a CAD model of the robot to estimate the "calibration" between the camera and the robot, which is the rigid body transformation between the camera and the robot arm's base. Evaluation results on the semantic segmentation component, end effector prediction component, and the final robot transformation. These results show that the calibration errors are relatively small.

**Issues:**

Please address my weakness concerns.

**Quality Of The Limitations Section:**

Limitations are addressed clearly

**Reviewer Expertise:**

2: The reviewer is willing to defend the evaluation, but it is quite likely that the reviewer did not understand central parts of the paper

**Robotics Focus:**

Highly relevant to robotics but no hardware experiments

**Strengths And Weaknesses:**

## Strengths
- Attempting to solve an important problem in robotics using learning
- Results appear quite good for the particular environment test scenarios

## Weaknesses
- W1: The use of learning for this problem is dubious. L294 The paper states "the system collects and labels its own data". Because the labels here are automatically collected, why is learning needed? If indeed the ground truth targets can be computed automatically, then why learn anything? Isn't this automatic labeling process already solving the problem? It's possible I'm missing some motivation or key understanding of the method. Can we consider a hypothetical experiment in which we replaced the learner's predictions with the process for automatically collecting the labels, and compare the performance, and explain why learning is useful to this setting?

- W2: The evaluation protocol is unclear. L294 The paper states "the system collects and labels its own data". Yet it's not clear how the labels for performance evaluation are collected. I think there needs to be a clear section on the difference between the labels used for training the system vs. the labels used for evaluating the system. Presumably the labels used for evaluating the system are not collected automatically (or at least they must be verified as correct by people in some way)? I think a simulation environment of the sensors would actually be beneficial here, because then ground truth performance metrics could be computed in a way that doesn't rely on human labels.

- W3: The evaluation is quite limited. My understanding is that it is evaluated on a single robot arm in a single lab, at different camera positions. To have strong evidence that the method is robust, more results either with a different robot arm or in a different lab setup location (or both) are needed.

- W4: The data collection protocol is unclear. L110 "The EE points are further obtained by transforming the EE bounding box using the calibration information and taking the points inside this box." I don't follow this explanation. What is the calibration information? Isn't the calibration information the final output (and therefore not yet available)? This prevents me from understanding how the EE semantic segmentation labels are computed. Also, "The CAD model of the robot along with joint positions and the calibration information can be used to get the arm and EE points, and consequently background points, as well. This would be needed when the robot arm is visible in the frame no matter what, to get the background points." Using the word "can" here (at the beginning of the sentence) is confusing, because it leads to multiple interpretations (1) One could devise a method to use the CAD model of the robot, *but we did not* (2) One could devise a method to use the CAD model of the robot, *and we did*. Please clarify.

- W5: The method is unclear. Figs 2 and 3 could be significantly improved by including visualizations of the output of each module. Then the reader won't have to mentally visualize each component. Furthermore, it's confusing to separate Fig 2 and Fig 4 across so many pages. I can't determine how the outputs of the module in Fig 2 are used as inputs somewhere in Fig 4. Is it the "Frame Predictions" box in Fig 4? This is ambiguous (they should be called the same thing in both figures)?. The box "Calibration Results" in Fig 4 is also ambiguous. What exactly are the calibration results? The "type" of the results needs to be stated. Is it the 3D rigid transformation from the a robot’s base and the camera?

**Summary Of Recommendation:**

I have significant doubts that may stem from misunderstanding. Overall, I found the method fairly difficult to understand.

---

> ### Author Response · Authors · 2022-08-18
> **Initial Rebuttal and Action Plan Related to Reviewer Comments**
>
> We thank the reviewer for their invaluable time and useful feedback.
>
> W1: The main motivation of our work is to handle the calibration changes during the lifetime of the robot. The calibration between the robot and the camera needs to be updated frequently due to multiple issues some of which are outlined in the intro. This is very prevalent in a lab setting with low cost robots, sensors, fixtures, and multiple users. Typically each user needs to perform calibration before working on their project. In our approach, this calibration is done once per generic camera location to collect the data. The automatic labelling is done with an existing calibration. Once the robot and the camera are out of calibration, we cannot collect automatic labels. Our method is used to handle the calibration automatically after training.
>
> W2: We perform manual calibration before collecting test data and apply automatic labeling afterwards. We agree that simulation would be very beneficial, however, one of our aims is to eliminate simulation usage all together. One issue with simulation is that the noise models for depth cameras are usually not realistic.
>
> W3: We agree that our results are lacking. We are in the process of performing more detailed evaluations and will upload the new results accordingly soon.
>
> W4: Calibration information for training is coming from manual tuning. This is only used during training and not available during deployment/inference.
>
> We did not use the robot CAD model to get background data as our setup did not require it.  However in cases where the robot cannot be easily removed from the background via subtraction, CAD models and ICP can be used. We will clarify this further in the text.
>
> W5: These are very valid points. We will improve the figures and their positioning. We will share the updated figures here.

---

> ### Author Response · Authors · 2022-08-28
> **Final updates on work**
>
> We shared our modified paper and the details as a comment to meta review. Please use https://openreview.net/forum?id=eI8CZ2s267o&noteId=iAa3zy_4SXN to view the modified paper and our comments.

---

### Meta-Review · Area_Chair_no1R · 2022-08-15

**Recommendation:** Accept (Poster)
**Confidence:** 4

**Metareview:**

Dear Authors,

Thank you for submitting your manuscript to CoRL. I'm happy to inform you that your paper has been accepted. We have completed the review of your manuscript and a summary is appended below.
- The responses from authors have addressed concerns of most reviewers.
- The reviewers have advised accepting your manuscript as a poster after improvement of the quality of the manuscript based on the comments.
- Please note it is crucial to incorporate all provided explanations by authors and recommended editing by reviewers into the final manuscript.

Regards,

---

> ### Author Response · Authors · 2022-08-28
> **Updates on our work**
>
> We would like to thank all of our reviewers for their time. We have made following updates on our work according to their comments and suggestions. We have modified our paper and marked all the additions with orange.
>
> - We have clarified data collection and method steps
> - We have added more detailed results
> - We have added an AR-tag based baseline
> - We have expanded our limitations and future work
>
> The only major point we did not address is comparing against other learning based methods. This is due to the setup (heavy reliance on simulation data and lack of point cloud in reference datasets, camera poses) and machine learning task discrepancies (e.g. predicting each joint location)
>
> We have exceeded the page limits but we did not spend effort on making the additions more concise, for now. We didn't want to change the existing text too much and as a result kept the new additions separate. However, we can have the new text blend and reformulate the tables so that we can fit in the allotted page limit for the camera ready version. We shared our modified paper as a file in this comment.